# Gut Microbiota and Immunotherapy for Alzheimer’s Disease

**DOI:** 10.3390/ijms232315230

**Published:** 2022-12-03

**Authors:** Chun-Ling Dai, Fei Liu, Khalid Iqbal, Cheng-Xin Gong

**Affiliations:** Department of Neurochemistry, Inge Grundke-Iqbal Research Floor, New York State Institute for Basic Research in Developmental Disabilities, Staten Island, New York, NY 10314, USA

**Keywords:** Alzheimer’s disease, gut microbiota, tau, amyloid β, immunotherapy

## Abstract

Alzheimer’s disease (AD) is a progressive neurodegenerative disorder that eventually leads to dementia and death of the patient. Currently, no effective treatment is available that can slow or halt the progression of the disease. The gut microbiota can modulate the host immune system in the peripheral and central nervous system through the microbiota–gut–brain axis. Growing evidence indicates that gut microbiota dysbiosis plays an important role in the pathogenesis of AD, and modulation of the gut microbiota may represent a new avenue for treating AD. Immunotherapy targeting Aβ and tau has emerged as the most promising disease-modifying therapy for the treatment of AD. However, the underlying mechanism of AD immunotherapy is not known. Importantly, preclinical and clinical studies have highlighted that the gut microbiota exerts a major influence on the efficacy of cancer immunotherapy. However, the role of the gut microbiota in AD immunotherapy has not been explored. We found that immunotherapy targeting tau can modulate the gut microbiota in an AD mouse model. In this article, we focused on the crosstalk between the gut microbiota, immunity, and AD immunotherapy. We speculate that modulation of the gut microbiota induced by AD immunotherapy may partially underlie the efficacy of the treatment.

## 1. Introduction

Alzheimer’s disease (AD) is a devastating, progressive neurodegenerative disease that impairs cognitive function, often combined with psychiatric symptoms such as mood, behavior, or personality changes, and eventually leads to dementia and death of the patient. The histopathological hallmarks of AD are the accumulation of extracellular senile plaques consisting of amyloid-β (Aβ) peptides [1] and of intracellular neurofibrillary tangles (NFTs), composed of aggregated abnormally hyperphosphorylated microtubule-associated protein tau [2], as well as neuroinflammation and neuronal and synaptic loss. AD progresses slowly, beginning 20 years or more before symptoms emerge [3]. There are two forms of AD: early onset and late-onset. Early onset AD is rare, representing less than 5% of all AD cases, some of which are caused by mutations in certain genes, such as amyloid precursor protein (APP), presenilin 1, or presenilin 2. It typically occurs during the fourth to sixth decades of life. Most AD cases are late-onset, in which symptoms become apparent after the mid 60s. The etiologies and mechanisms of late-onset, sporadic AD are still not clearly understood. Sporadic AD has multiple etiologies, including genetic risks, epigenetic and metabolic factors, and environmental insults [4]. Electromagnetic radiation, to which human beings are increasingly exposed in modern society, appears to affect the brain and might also contribute to the development of sporadic AD [5]. AD is the fifth-leading cause of death in individuals older than 65 years of age [3] and exerts a huge psychological, social, and economic burden on modern society. The cost for treating and caring for patients with AD and other dementias in 2022 may reach $321 billion in the United States alone [3]. Therefore, it is urgent to develop therapeutic treatments that can stop and/or delay the onset and progression of AD.

The gut microbiota, comprising trillions of microorganisms (bacteria and fungi) and viruses, plays a pivotal role in the development, digestion, behavior, and immune system of the host [6,7,8]. Importantly, recent studies indicate that the gut microbiota can also regulate brain function actively through the microbiota–gut–brain (MGB) axis, which is a bidirectional communication network between the central nervous system (CNS) and the gastrointestinal tract [9,10,11,12,13]. Gut microbiota dysbiosis has been reported to play a significant role in the pathogenesis of AD (reviewed in refs. [12,14,15,16]).

No effective treatment is available for AD. The commonly used AD drugs include acetylcholinesterase inhibitors (donepezil, galantamine, and rivastigmine) for mild to moderate AD and the NMDA receptor antagonist memantine (individually or in the combination with donepezil) for moderate to severe AD [17]. Unfortunately, these drugs only temporarily treat Alzheimer’s symptoms but do not change the underlying brain changes of AD or alter the course of the disease. Therefore, development of disease-modifying therapeutics is urgent for combating AD. Currently, 143 agents are in 172 clinical trials for AD. Disease-modifying therapies (119 agents) represent 83.2% of the total number of agents in AD clinical trials [18]. Given the critical role of Aβ and tau in the pathogenesis and progression of AD, immunotherapy targeting Aβ and/or tau is currently the major focus for the development of disease-modifying therapy for AD. This approach harnesses the immune system to clear the pathological Aβ or tau protein and block the propagation of Aβ or tau pathologies among the neurons. Currently, 42 AD immunotherapy clinical trials are underway, which include 2 active immunizations and 15 monoclonal antibodies (mAbs) targeting various forms of Aβ or tau (Table 1, Table 2 and Table 3). One Aβ mAb, aducanumab (marketed as Aduhelm), received FDA-accelerated approval in June 2021 for mild cognitive impairment and mild dementia due to AD [3]. Encouraging results from a phase II clinical trial with another Aβ mAb, lecanemab, were recently reported (https://www.eisai.com/news/2022/news202271.html (accessed on 25 October 2022)), and an application for its accelerated FDA approval for treating early stage AD is currently under review. However, the exact therapeutic efficacies and mechanisms of these AD immunotherapeutic agents require further investigation.

The human immune system and the microbiota co-evolve, and their balanced relationship is based on the crosstalk between the two systems throughout life. Dynamic interactions between the gut microbiota and the host’s innate and adaptive immune systems play an important role in the modulation of host immunity homeostasis [19]. The gut microbiota metabolizes proteins and complex carbohydrates, synthesizes vitamins, and produces an enormous number of metabolic products that can mediate crosstalk between gut epithelial and immune cells. Gut microbiota dysbiosis in the host can dysregulate immune response and cause susceptibility to infections, hypersensitivity reactions, autoimmunity, chronic inflammation, and cancer [20,21,22]. Importantly, gut microbiota dysbiosis can impact the immune response to immunotherapy [21]. Accumulating evidence indicates that gut microbiota dysbiosis affects the therapeutic effects of immune checkpoint inhibitors, and restoration of the gut microbiota could increase the immunotherapy response in cancer patients [23,24,25,26,27]. The therapeutic efficacy of AD immunotherapy may also depend on the normal immune system. Immune dysregulation in the peripheral nervous system (PNS) and CNS was reported in AD [28], and dysregulation of the gut microbiota has been observed both in AD patients [29,30,31] and in mouse models of AD [32,33,34,35]. Similarly, immune dysregulation and gut microbiota dysbiosis were also reported in other neurodegenerative diseases, such as Parkinson’s disease (PD), amyotrophic lateral sclerosis, and multiple sclerosis [36,37]. Alpha-synuclein (α-Syn) protein production and aggregation are the major neuropathologic feature of PD, and α-Syn might serve as a chemoattractant that enhances the local immune response in the gut. Accumulation of α-Syn has been observed after bacterial and viral infection in the gut, and thus intestinal inflammation has been proposed as an environmental link to neurodegeneration, with disease beginning in the gut and spreading to the brain via the vagus nerve [38]. To date, very few data about the gut microbiota have been reported from clinical trials of AD immunotherapy. Given the significant impact of the gut microbiota on cancer immunotherapy, it is important to consider gut microbiota dysbiosis of trial participants and to include studies of the gut microbiota in AD immunotherapy clinical trials.

In this article, we addressed relationships among the gut microbiota, AD, and AD immunotherapy, especially whether AD immunotherapy can exert its therapeutic efficacy through modulation of the gut microbiota.

## 2. Gut Microbiota and the Microbiota–Gut–Brain Axis

The human microbiota, composed of a variety of microorganisms, such as bacteria, fungi, archaea, protozoa, and viruses, reside on the surface of our body’s epithelial barrier [39,40]. The human body contains about 100 trillion bacteria in the intestines, which is 10 times greater than the total number of human cells in the body. The human microbiome is composed of more than 5000 strains of microbes and more than 1000 kinds of microflora [41,42,43]. The bacteria, mainly anaerobic bacteria, dominate this environment, and the others, including viruses, protozoa, archaea and fungi, are also involved in this environment [44,45]. The microbiome is mainly defined by two bacterial phylotypes, *Bacteroidetes* and *Firmicutes*, and the number of *Proteobacteria*, *Actinomyces*, *Fusobacterium*, and *Verrucomicrobia* is relatively small [46]. The well-known gut bacteria *Escherichia* coli constitutes only about 0.1% of gut microbiota [47]. The composition of the gut microbiota is age-sensitive, with dramatic differences during infancy, adolescence, adulthood, and senescence [45]. It also depends on many factors such as genetics, stress, mode of birth, diet and exercise, medication, and the environment [15,48,49]. The abundant and diverse microbial composition plays critical roles in the maintenance of human health. The gut microbiota constitutes the intestinal barrier, promotes the continuous existence of gut microorganisms, stimulates intestinal epithelial cell regeneration, and produces mucus and nourishes mucosa by producing short-chain fatty acids (SCFAs) [50,51]. It also modulates the immune system by stimulating the innate immune system in the early stage of life, assists in the maturation of intestinal-related lymphoid tissue, and inspires acquired immunity by stimulating local and systemic immune responses [43,52]. In addition, the gut microbiota involved in intestinal synthesis and metabolism of certain nutrients, hormones, and vitamins plays an important role in drug and poison removal and protects the host from invasion by pathogens [15,39]. Under normal physiological conditions, the gut microbiota continues to stimulate the immune system, leading to a state of “low degree of physiological inflammation,” which is a rapid and effective mechanism for defending against pathogens [43,53].

Recent studies indicate that the gut microbiota can regulate brain function actively through the MGB axis [9,10,11], which consists of the bidirectional communication network between the gastrointestinal system and the CNS. The primary function of the MGB axis is to monitor and integrate intestinal functions and to link, through immune and neuro-endocrine mediators, the emotional and cognitive centers of the brain to peripheral intestinal mechanisms, such as immune activation, intestinal permeability, enteric reflex, and entero-endocrine signaling [14,43,54]. Multiple pathways are involved in the communication between the gut and the brain [55].

The vagus nerve is a major modulatory pathway of the MGB axis. It is composed of 80% afferent and 20% efferent fibers and can sense the microbiota metabolites (such as serotonin and glutamate) through its afferents, transferring this gut information to the CNS, where it is integrated into the central autonomic network, and then generating an adapted or inappropriate response to modulate the function of gut microbiota directly and indirectly [56,57]. A reduction in vagal tone reflecting dysautonomia has been shown in irritable bowel syndrome and inflammatory bowel disease [58], characterized by a leaky gut and dysbiosis [55,59]. Bacterial metabolites, such as SCFAs, bioactive peptides, and the modulation of transmitters, such as serotonin and acetylcholine, play a crucial role in the network of the MGB axis. SCFAs have immunomodulatory properties and can interact with nerve cells by stimulating the sympathetic branch of the autonomic nervous system. Furthermore, microbiota-derived SCFAs can cross the blood–brain barrier (BBB) and regulate microglia homoeostasis, which is required for proper brain development and brain tissue homoeostasis and is involved in behavior modulation. Disruptions of SCFA metabolism have been implicated in the development of autism through the disruption of microglial communication and function [55,60]. SCFAs also regulate the release of gut peptides from enteroendocrine cells and the synthesis of gut-derived serotonin from enterochromaffin cells, both of which in turn affect gut–brain hormonal communication [55]. The activation of the immune system is involved in the network of the MGB axis. Inflammation metabolism within the gastrointestinal tract is influenced by the gut microbiome, principally via the immune systems’ release of cytokines, such as IL-10 and IL-6, and other cellular communication mediators, such as interferon-*γ*, during dysbiosis. In irritable bowel syndrome, abnormal microbiota populations activate mucosal innate immune responses, which increase gut epithelial permeability, activate gut pain sensory pathways, and dysregulate the enteric nervous system. Disruptions in the gut–brain axis affect intestinal motility and secretion, contribute to visceral hypersensitivity, and lead to cellular alterations of the entero-endocrine and immune systems [55]. The crosstalk between the components of the MGB axis can affect, through the secretion of cortisol by the hypothalamic–pituitary–adrenal axis during stress, intestinal motility, integrity, and mucus production, leading to changes in gut microbiota composition. These alterations, in turn, may affect the CNS through the modulation of stress hormones [14,61].

In the network of the MGB axis, inputs from the CNS can modify gut functions, while inputs from the gut to the CNS can modulate specific symptoms [62]. Alterations of these bidirectional communications may contribute to neuroinflammation and the pathogenesis of CNS disorders. Growing evidence indicates that gut microbiota dysbiosis is closely associated with the pathogenesis of CNS diseases such as depression, Parkinson’s disease, and AD [63,64,65].

## 3. Gut Microbiota and AD

The aging process is accompanied by the occurrence and development of inflammation. Since the elderly usually have a variety of comorbidities, changes in diet and exercise habits, and other changes associated with gastrointestinal activity that affects the gut microbiota, aging has a strong impact on gut microbiota composition, favoring the development of pro-inflammatory bacteria (such as *Bacillus fragilis*, *Faecalibacterium prausnitzii*, *Eubacterium rectale*, *Eubacterium hallii*, and *Bacteroides fragilis*) to the detriment of anti-inflammatory bacteria. These age-related gut microbiota changes can induce local systemic inflammation, leading to enhanced permeability of the gastrointestinal tract, and promote BBB impairment and neuroinflammation. Various gut microbes (such as *Actinobacteria*, *Bacteroidetes*, *E. coli*, *Firmicutes*, *Proteobacteria*, *Tenericutes*, and *Verrucomicrobia*) and their metabolites (such as SCFAs) can play a significant role in the pathogenesis of AD via modulation of various pathophysiological processes involved in AD pathogenesis, such as neuroinflammation and other inflammatory processes, amyloid deposition, and BDNF and NMDA signaling [11,66]. Gut microbiota alterations or infection with toxic bacteria or their secretory products into the brain may contribute to the development of AD by triggering or accelerating neuroinflammatory and neurodegenerative process [67,68]. Gut microbiome-derived lipopolysaccharides (LPSs) have been found within human brain cells both during advanced aging and in AD brain [69,70]. Microbiome derived *E. coli* LPSs and *Bacillus fragilis* LPSs are found to be associated with the hippocampal CA1 and neocortical regions in AD brain [69,70]. Growing evidence has revealed that the gut microbiota composition is altered both in AD patients [29,30,31] and in AD mouse models [32,33,34,35]. A new drug GV−971, which was recently approved by China FDA for treating AD, might work through modulation of the gut microbiota in patients [71,72]. Thus, modulation of the gut microbiota dysbiosis may open a new avenue for treating AD. Recently, it was reported that restoration of the gut microbiota via transfer of healthy gut microbiota into AD mouse models reduced amyloid and tau pathologies and cognitive deficits [73,74], and similarly, such treatments improved cognitive function in AD patients [75,76].

Among environmental factors that contribute to the development of AD is diet [77]. Diet can exert significant influence on gut microbiota composition and inflammation [78]. A diet with high fruit and vegetable consumption, moderate consumption of poultry, fish, eggs, and dairy, and low consumption of red meat and processed foods may protect against chronic inflammation and related diseases, including AD [79]. Considering the significant role of gut microbiota in the development of AD, gut microbiota may partially underlie the diet-associated risk for AD.

## 4. Immunotherapy for AD

The innate immune system is a well-conserved host defense system and is responsible for elimination of any challenge rapidly and non-specifically. It generates non-specific inflammatory responses as a necessary part of the defense response to these aggressions, directed to extra- and intracellular pathogens. The innate immune system is composed of neutrophils, monocytes/macrophages, dendritic cells, and natural killer (NK) cells in the peripheral system. Microglia and astrocytes are the predominant innate immune cells in the CNS and are also involved in stimulating adaptive immunity. As the major immunological effector of the innate immune system in the CNS, microglia dynamically survey the environment and thus play a crucial role in CNS tissue maintenance, injury response, and pathogen defense [80,81,82,83]. Microglia also participate in the developmental sculpting of neural circuits by engulfment and removal of unwanted neurons and synapses [84,85].

Clinical studies suggest that the peripheral and central immune system are dysregulated in AD and are related to cognitive function and clinical status [86]. Genome-wide association studies (GWAS) and pathway analyses emphasize that the innate immune system and neuroinflammation play important roles in the pathogenesis and progression of AD [87,88,89,90]. Microglial cells can be activated during systemic infections without the integrity of the BBB being compromised. Some regions of the brain have no BBB, and the response to circulating pathogens at these sites is similar to that in most systemic organs. The circumventricular organs, which include the organum vasculosum of the lamina terminalis, the subfornical organ, the median eminence, and the area postrema, are strategically positioned to detect pathogen-associated molecular patterns in the blood. These regions of the brain have a rich vascular plexus with a specialized arrangement of blood vessels, and the junctions between capillary endothelial cells at these regions are not tight, allowing for the diffusion of large molecules from the capillaries into the CNS. Although they are not considered circumventricular organs, the choroid plexus and leptomeninges are also highly vascularized, and microglial cells in these regions are rapidly activated by circulating pathogens. These organs are characterized by a high density of small neurons, astrocytes, macrophages, and microglial cells. Rapid innate immune responses to systemic infection are initiated at these regions, followed by the progressive activation of resident microglial cells in the brain parenchyma [88,91,92]. Additionally, immune cells may travel to and from the brain as a result of altered permeability of the BBB in AD patients. Importantly, gut microbiota dysbiosis may lead to a systemic inflammatory state, which can impede the function of the brain cells, including the activation of microglia at the origin of neuroinflammation. Then, a vicious circle is initiated between the brain and the gut that is facilitated by increased BBB permeability [93,94]. These processes imply that peripheral innate immune cells participate in the pathogenesis and progression of the disease. Activation of peripheral innate immune cells may represent early biomarkers of brain pathology, and targeting the innate immune system may present a strategy to modify brain disease progression [95].

Harnessing the immune system to prevent or remove the Aβ and tau aggregates is believed to be a promising disease-modifying approach for combatting AD. Active immunotherapy and passive immunotherapy against Aβ or tau have been the most widely studied therapeutic approaches against AD over the past two decades. Active immunization involves administration of a vaccine containing Aβ or tau antigens with other stimuli designed to induce an immune response that generates antibodies in the recipient. In passive immunization, mAbs are administered by intravenous infusions or subcutaneous injection [96]. Active and passive immunotherapy both have advantages and disadvantages. The advantages of active immunization are the generation of long-term antibody response after a small number of vaccinations and the production of polyclonal antibodies with multiple specificities against the antigen. A potential disadvantage of active vaccination is the variability in the antibody response across patients, which may be especially problematic in the context of AD because of age-related reductions in the immune competency of elderly patients. The senescent immune system is less likely to generate therapeutically adequate titers of antibodies in response to vaccination and is more likely to develop autoimmune side effects. Additionally, side effects may be persistent over the long term if adverse effects develop after active vaccination. The potential advantages of passive immunotherapy include the reproducible delivery of a known amount of therapeutic antibodies to the patient and the rapid clearance of those antibodies if side effects develop. A disadvantage is the requirement for repeated infusions of antibodies over time [96]. Therefore, passive immunization has been developed much faster than active immunization. Among 42 ongoing clinical trials for AD immunotherapy, only 2 of them are for active immunization (Table 1 and Table 2).

In both active and passive immunization, the antibodies, at first located in the peripheral circulating system, are required to cross the BBB to reach the CNS. The access routes for immunoglobulins into the CNS have not been clearly identified yet, but the lymphatic system, passive diffusion, and perivascular spaces within the CNS, in which the BBB is leaky, may contribute to the transport of antibodies into the CNS. However, only a small fraction of antibodies (approximately 0.1%) in the peripheral circulation can enter the CNS because of the absence of active transport systems for antibodies, the presence of receptors (such as the neonatal Fc receptor) acting as a pump to remove antibodies from the CNS, and probably other yet-unidentified clearance mechanisms [96]. Thus, whether and how such small amounts of antibodies in the brain can exert therapeutic efficacy remains to be determined.

Research evidence suggests that Aβ immunotherapy may exert therapeutic efficacy through multiple mechanisms both centrally and peripherally. First, antibodies in peripheral circulation may cause a change in Aβ equilibrium between the CNS and plasma. It has been reported that long-term peripheral administration of mAb m 266 to PDAPP transgenic mice results in a rapid 1000-fold increase in plasma Aβ, with reduction of Aβ deposition, without binding to Aβ deposits in the brain [97]. Aβ antibodies in the circulation can sequester plasma Aβ and disrupt the Aβ equilibrium between the CNS and peripheral blood, resulting in a driving force for movement of Aβ out of the brain into the periphery to be degraded [96,98,99,100,101]. Second, antibodies in the CNS may bind to soluble forms of Aβ and increase its clearance or bind to the deposited amyloid plaque and facilitate microglial cells to clear plaques through Fc receptor-mediated phagocytosis and subsequent peptide degradation [96,98,102] or non-Fc-mediated clearance of Aβ plaques in brain [103,104]. Third, the epitope EFRH, corresponding to amino acids 3–6 of human Aβ, acts as a regulatory site controlling both the formation and disaggregation process of Aβ amyloid [105]. Antibodies against the N terminus of Aβ, including this epitope, may thus affect the dynamics of the entire Aβ molecule, prevent self-aggregation, and enable re-solubilization of already-formed aggregates to nontoxic, normal components [105,106,107,108]. Finally, antibodies might even be internalized in cells or enter the synaptic clefts between neurons, with the potential to interfere with cell-to-cell transmission of Aβ and its aggregation [96].

Immunotherapy targeting tau has recently emerged as a hot subject in the field of AD therapeutics research. This approach may be more promising because tau pathology correlates well with dementia symptoms [109,110]. Clearance of extracellular tau may be the primary mechanism of tau immunotherapy. Tau is predominantly a cytoplasmic protein that stabilizes microtubules [111,112]. Tau may be actively released into the extracellular space under physiological conditions [112,113] and be passively released during neurodegeneration. Neuronal activity can regulate tau release, and high neuronal activity promotes tau release both in cultured neurons in vitro and in mouse brain in vivo [114,115,116,117]. Tau level in the cerebrospinal fluid (CSF) increases in aged healthy individuals and in AD patients [118,119,120,121,122]. Although the exact function of extracellular tau remains elusive, studies indicate that a high level of CSF tau is associated with faster clinical progression of AD [123]. Thus, reduction of extracellular tau with tau antibodies may lessen the neuronal dysfunction induced by extracellular tau, which may in turn reduce the release of tau into extracellular space. By using a triple transgenic AD mouse model (3xTg-AD), we found that intravenous administration of a tau antibody reduces tau level and ameliorates tau pathology in mouse brains in a dose-dependent manner [124].

Some tau antibodies may enter the neuron and bind to intracellular pathological tau to promote tau clearance. It has been reported that tau antibodies can be taken up by neurons and promote intracellular sequestration and clearance of tau [125,126,127,128,129,130,131]. Fc receptor–mediated endocytosis and the endosome-autophagosome-lysosome system are believed to play a critical role in antibody-mediated clearance of tau pathology. Intraneuronal antibodies may localize in the endosomal-autophagosome-lysosome system and promote tau clearance by degrading tau aggregates. Importantly, antibody uptake into neurons has been shown to be a prerequisite for acute tau clearance. This intracellular interaction may sequester the tau protein, preventing its release into the extracellular space and subsequent spread in the brain [132].

Tau immunotherapy may also target the transcellular propagation of pathological tau. Tau pathology is well documented to propagate in a predictable pattern [133,134]. Studies of autopsied brains indicate that neurofibrillary pathology in the brains of AD patients starts in the entorhinal/perirhinal cortex and spreads anatomically in a defined pattern to the limbic system and eventually to the isocortex [135,136]. A stereotypical pattern of tau pathology progression similar to that in AD has been shown experimentally in different mouse models [137,138,139,140,141]. The abnormally hyperphosphorylated/oligomeric tau released into the extracellular space from the affected neurons is suspected to serve as seeds for the spread of tau pathology by the ingesting cells [137,142]. Therefore, tau immunotherapy could provide a potential therapeutic opportunity by clearance of extracellular tau that is involved in the spread of the pathology in AD and other tauopathies. Treatment with tau antibodies can block the seeding activity in vitro and inhibit the spread of tau pathology in tau-transgenic mice [143,144,145]. We also found that immunization with tau antibody can inhibit the seeding of AD hyperphosphorylated tau (AD p-tau) and block the propagation of pathological tau templated by AD p-tau [146].

Both active and passive immunotherapy may induce over-activation of the innate and adaptive immune systems, resulting in adverse side effects. As with other vaccines, the adverse effects of AD immunotherapy are likely very mild. The common adverse effect of AD immunotherapy is infusion reactions, which can be easily managed clinically. Other rare adverse events that have been reported from available clinical trials of AD immunotherapy include amyloid-related imaging abnormalities, cerebral microhemorrhage, dizziness, headaches, nausea, rashes, and diarrhea [147,148,149,150]. A recent meta-analysis found no significantly increased incidences of these rare adverse events in the passive immunization groups as compared to the placebo groups [151]. Immune responses and adverse reactions are generally hard to predict for active immunotherapy as compared with passive immunotherapy, especially in senior people. This is probably why much more passive immunotherapies for AD are currently under development.

## 5. Can AD Immunotherapy Act through Modulation of the Gut Microbiota?

Recent studies suggest that the gut microbiota may play an important role in the pathogenesis of AD [152]. Germ-free (GF) 3xTg-AD mice display reduced microglia activation and Aβ and tau pathologies as compared with specific-pathogen-free animals [153]. However, GF mice showed significantly increased brain Aβ levels after transplantation of fecal samples from AD mice [94] or AD patients [153]. Importantly, restoration of the gut microbiota via fecal microbiota transplantation could reduce tau and Aβ pathologies in mouse models [73,74]. These findings indicate that manipulation of the gut microbiota can influence Aβ deposition and neuroplasticity processes [154]. Given that immunization is expected to modulate the host immune system and the gut microbiota, we speculated that AD immunization could also exert its efficacy through modulation of the gut microbiota. This speculation is supported by our findings that immunization of 3xTg-AD mice with a tau antibody, which does not recognize Aβ or Aβ plaques, also decreases Aβ pathology in the mouse brain [124,146]. Reduction of Aβ pathology after tau immunotherapy was also reported in an independent study [155]. This hypothesis may explain why benefits, including reduction of Aβ and tau pathologies, were sometimes observed in AD mouse models treated with non-specific mouse immunoglobulins [156,157,158].

To test this hypothesis, we collected fecal samples from 3xTg-AD mice and wild-type control mice both before and after immunization with tau antibody 43D weekly for six weeks and studied the gut microbiota composition. We found significant changes in the gut microbiota in 3xTg-AD mice, such as reduced proportions of the phylum *Cyanobacteria* and the order *Turicibacterales*, and increased proportion of the class *Gammaproteobacteria* (unpublished observations). Cyanobacterial toxin metabolites play an important role in the gastrointestinal tract and mucosal innate immune system [159]. *Gammaproteobacteria* normally represent only a very small proportion of the healthy adult gut microbiome, but they are important for immune patterning and maintenance of mucin integrity [47]. As a major contributor to LPS production, the elevation of *Gammaproteobacteria* is positively associated with metabolic disorders and inflammation [160], which may impair intestinal integrity and cause chronic intestinal inflammation [161,162]. We found that immunization with tau antibody 43D restored the proportions of *Gammaproteobacteria* and *Turicibacterales* in the gut microbiota of 3xTg-AD mice (unpublished observations). Thus, it is possible that the therapeutic efficacy of tau antibody 43D [124,146] may attribute, at least partially, to its action to modulate the gut microbiota in these mice. Taken together, AD immunotherapy is likely to exert its therapeutic efficacy through multiple pathways, including modulation of the gut microbiota.

## 6. Does the Gut Microbiota Affect the Therapeutic Effects of AD Immunotherapy?

The gut microbiota is well known to modulate the host’s immune system both locally and systemically and to affect the therapeutic efficacy of cancer immunotherapy [40,163]. Gut microbiota dysbiosis can lead to primary resistance to immune checkpoint inhibitors (ICIs), and the use of antibiotics inhibits the clinical benefit of ICIs in patients with advanced cancer [27]. Fecal microbiota transplantation (FMT) from cancer patients who responded to ICIs into germ-free or antibiotic-treated mice ameliorates the antitumor effects of PD-1 blockade, whereas FMT from nonresponding patients fails to show the improvement of antitumor activity [24,25,26,27]. Importantly, FMT from responder patients overcomes resistance to anti-PD-1 therapy in melanoma patients [164]. These findings in cancer immunotherapy provide new insights into the relationship between the gut microbiota and AD immunotherapy. However, the completed AD immunotherapy clinical trials did not include studies of the gut microbiota of the trial subjects. The gut microbiota plays an important role in the pathogenesis of AD, and alterations of the gut microbiota are found in AD patients [29,30,31]. Importantly, transfer of healthy gut microbiota reduces amyloid and tau pathologies and cognitive deficits in AD mouse models [73,74] and improves cognitive function in AD patients [76,165]. It would not be surprising if gut microbiota dysbiosis in AD patients could negatively impact the clinical outcomes of AD immunotherapy. On the other hand, promoting healthy gut microbiota, which could be achieved by healthy diet [166], may benefit AD immunotherapy. Thus, including gut microbiota studies and stratifying clinical trial participants according to their gut microbiota status in future AD immunotherapy clinical trials will likely provide more insights. Restoring gut microbiota homeostasis for clinical trials of AD immunotherapy might have a better chance to achieve the beneficial outcomes of AD immunotherapy.

## 7. Conclusions and Perspective

AD immunotherapy targeting Aβ and/or tau represents one of the most promising disease-modifying therapies for AD. In addition to directly binding to their specific antigens, AD immunotherapy might exert its therapeutic efficacy through a novel mechanism of modulating the gut microbiota. Future studies in germ-free, antibiotic-treated, and fecal transplantation models will be required to confirm this hypothesis and to explore the role of the gut microbiota in AD immunotherapy.

Aducanumab (Aduhelm) is the first AD immunotherapy approved by the FDA through the accelerated approval pathway. Although it can effectively reduce the amyloid pathology burden in the brains of patients, its clinical efficacy is very weak and debatable [3,167]. Analyses of the gut microbiota data of the trial participants, if available, may help elucidate the discrepancy in efficacies between the brain pathology and clinical outcomes by this immunotherapy. Because different antibodies might have different impacts on gut microbiota, monitoring gut microbiota in AD immunotherapy studies can help understand the mechanism of the treatment and help explain any discrepancies between changes in brain pathologies, which can be detected by brain imaging, and in cognitive function. Including gut microbiota studies in the promising lecanemab trial (clinical trial #s NCT04468659, NCT03887455, and NCT01767311) and other AD immunotherapy clinical trials should be seriously considered.

The human microbiome is composed of more than 5000 strains of microbes and more than 1000 kinds of microflora [41,42,43]. The huge diversities of microorganisms in the gut pose a challenge to investigate the role of gut microbiota in progression and in immunotherapy in AD. The use of antibiotics for patients with bacterial infections, which can change gut microbiota significantly, also presents another challenge to this type of investigation. Nevertheless, determining the gut microbiota of an AD patient might offer additional information beyond what the commonly used AD biomarkers and brain imaging do for projecting the progression of the disease and for optimizing the personalized treatment for the patient.

AD is a multifactorial disease involving several etiological factors and pathophysiological pathways [4]. These etiological factors include genetic (e.g., mutations in presenilin and APP genes, ApoE-e4 allele, etc.), epigenetic (e.g., DNA methylation, RNA interference, histone modification, etc.), metabolic (e.g., amyloid β accumulation, insulin resistance, oxidative stress, diet, etc.), and environmental factors (e.g., education, social engagement, traumatic brain injury, etc.). Multi-target and combinatorial therapy are thus proposed to be more effective than single-target therapy for AD [168,169,170,171]. A combination therapy using donepezil and memantine has been approved by FDA for the treatment of moderate to severe AD [168]. Given the critical roles of Aβ and tau pathologies in the pathogenesis and development of AD, a phase III clinical trial targeting both Aβ protofibrils with lecanemab and tau with E2814 is ongoing (clinical trial # NCT05269394). Considering the possible role of gut microbiota in the development of AD, restoration of gut microbiota should be considered as a potential approach for treating AD. Given the potential influence of gut microbiota on the treatment of AD, future studies will establish whether a combination of restoration of gut microbiota with other available AD treatments can further improve the overall therapeutic benefits to patients.

## Figures and Tables

**Table 1 ijms-23-15230-t001:** Current ongoing clinical trials for Aβ immunotherapy (Clinicaltrials.gov, as of 1 October 2022).

Agent	Immunization	Target	Phase	Study Population	Clinical Trial #	Status	Sponsor(s)
ABvac40	Active	Aβ (33–40)	Phase II	Amnestic mild cognitive impairment or very mild AD	NCT03461276	Active, not recruiting	Araclon Biotech S.L.
Aducanumab (BIIB037)	Passive	Aβ (3–7)	Phase III	Early AD	NCT04241068	Active, not recruiting	Biogen
			Phase IV	Early AD	NCT05310071	Recruiting	Biogen
			Phase I	Mild cognitive impairment or mild AD	NCT05469009	Recruiting	Ali Rezai, InSightec
Gantenerumab (RO4909832)	Passive	Aβ (2–11 and 18–27)	Phase II	Early AD	NCT04592341	Active, not recruiting	Roche
			Phase III	Early AD	NCT03443973	Active, not recruiting	Roche
			Phase III	Early AD	NCT03444870	Recruiting	Roche
			Phase III	Early AD	NCT04339413	Active, not recruiting	Roche
			Phase III	With risk for AD or early AD	NCT05256134	Recruiting	Roche
			Phase III	Prodromal to mild AD	NCT04374253	Recruiting	Roche
			Phase III	With risk for or with early AD caused by a genetic mutation	NCT05552157	Not yet recruiting	WUSM, Roche, AA, NIA Genentech, Inc.
Donanemab (LY3002813)	Passive	Pyrogluta-mate Aβ (p3–7)	Phase I	Healthy	NCT05567159	Not yet recruiting	Eli Lilly
			Phase I	Healthy Chinese participants	NCT05533411	Not yet recruiting	Eli Lilly
			Phase III	Early symptomatic AD	NCT04437511	Active, not recruiting	Eli Lilly
			Phase III	Preclinical AD	NCT05026866	Recruiting	Eli Lilly
			Phase II	Symptomatic AD	NCT04640077	Active, not recruiting	Eli Lilly
			Phase III	Early symptomatic AD	NCT05508789	Not yet recruiting	Eli Lilly
Lecanemab (BAN2401)	Passive	Aβ protofibrils (1–16)	Phase I	Healthy	NCT05533801	Not yet recruiting	Eisai.
			Phase III	Preclinical AD	NCT04468659	Recruiting	Eisai. ACTC, Biogen, NIA
			Phase III	Early AD	NCT03887455	Active, not recruiting	Eisai. Biogen
			Phase II	Early AD	NCT01767311	Active, not recruiting	Eisai. Biogen
Solanezumab (LY2062430)	Passive	Aβ protofibrils (1–16)	Phase III	With risk for memory loss	NCT02008357	Active, not recruiting	Eli Lilly, ATRI
RO7126209 *	Passive	Aβ fibrils	Phase I/II	Prodromal or mild to moderate AD	NCT04639050	Recruiting	Roche
Crenezumab (MABT5102A)	Passive	Soluble Aβ oligomers (13–24)	Phase II	Preclinical autosomal dominant AD with PSEN1 E280A mutation	NCT01998841	Active, not recruiting	Genentech, Inc. NIA, Banner Alzheimer’s Institute
SHR−1707	Passive	Aβ	Phase I	Healthy young adult and elderly	NCT04973189	Recruiting	Shanghai Hengrui Pharmaceutical Co.
LY3372993	Passive	Pyrogluta-mated form of Aβ	Phase I	Healthy and AD	NCT04451408	Recruiting	Eli Lilly
			Phase III	Early AD	NCT05463731	Recruiting	Eli Lilly
ACU193	Passive	Soluble Aβ oligomers	Phase I	MCI or mild AD	NCT04931459	Recruiting	Acumen Pharmaceuticals, NIA

*: RO7126209 is a new version of gantenerumab, engineered to cross the BBB more easily using a “brain shuttle” technology. AA, Alzheimer’s Association; ACTC, Alzheimer’s Clinical Trials Consortium; ATRI: Alzheimer’s Therapeutic Research Institute; NIA: National Institute on Aging; WUSM, Washington University School of Medicine.

**Table 2 ijms-23-15230-t002:** Current ongoing clinical trials for tau immunotherapy (Clinicaltrials.gov, as of 1 October 2022).

Agent	Immunization	Target	Phase	Study Population	Clinical Trial #	Status	Sponsor(s)
ACI−35	Active	Tau 393–408 (pS396/S404)	Phase II	Early AD	NCT04445831	Active, not recruiting	AC Immune; Janssen
Bepranemab (UCB0107)	Passive	Tau (235–250)	Phase II	Mild cognitive impairment or mild AD	NCT04867616	Active, not recruiting	UCB Biopharma SRL
E2814	Passive	Tau (273–291, 296–314)	Phase II	Mild to moderate AD	NCT04971733	Active, not recruiting	Eisai.
			Phase I	Healthy	NCT04231513	Recruiting	Eisai.
JNJ−63733657	Passive	Tau204–225 (pTau212/217)	Phase I	Healthy Chinese participants	NCT05407818	Recruiting	Janssen
Semorinemab (RO7105705)	Passive	Tau (2–24)	Phase II	Moderate AD	NCT03828747	Active, not recruiting	Genentech, Inc.
Lu AF87908	Passive	Tau386–408 (pS396/S404)	Phase IPhase II	Healthy and AD Early AD	NCT04149860 NCT04619420	Recruiting Recruiting	H. Lundbeck A/S Janssen

**Table 3 ijms-23-15230-t003:** Current ongoing clinical trials for the combination of Aβ and/or tau immunotherapy (Clinicaltrials.gov, as of 1 October 2022).

Agents	Immunization	Target	Phase	Study Population	Clinical Trial #	Status	Sponsor(s)
Gantenerumab & Solanezumab	Passive	Aβ (2–11, 18–27, and 16–26)	Phase III	With risk for or with early onset AD caused by genetic mutation	NCT01760005	Recruiting	WUSM; Eli Lilly, Roche; AA; NIA; Avid Radiopharmaceuticals, AMP
Donanemab & Aducanumab	Passive	p3–7 and Aβ (3–7)	Phase III	Early symptomatic AD	NCT05108922	Active, not recruiting	Eli Lilly
Lecanemab & E2814	Passive	Aβ protofibrils (1–16) and Tau (273–291, 296–314)	Phase III	Early onset AD caused by genetic mutation	NCT05269394	Recruiting	WUSM, NIA, AMP, Eisai, AA

AA, Alzheimer’s Association; AMP, Accelerating Medicines Partnership; NIA, National Institute on Aging; WUSM, Washington University School of Medicine.

## Data Availability

Not applicable.

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
