# Peer review of "Gut Microbiota and Immunotherapy for Alzheimer’s Disease"

_ijms, 2022, doi:10.3390/ijms232315230_

Round 1
Reviewer 1 Report
The manuscript is very interesting, well organized and has a rationale.
Perhaps a graphical abstract would be appropriate, in this way the conclusions are more immediate.
Perhaps these recent manuscripts could improve the manuscript by highlighting the role of the microbiota.
AÄŸagündüz D, Gençer Bingöl F, Çelik E, Cemali Ö, Özenir Ç, ÖzoÄŸul F, Capasso R. Recent developments in the probiotics as live biotherapeutic products (LBPs) as modulators of gut brain axis related neurological conditions. J Transl Med.
2022 Oct 8; 20 (1): 460.
AÄŸagündüz D, Kocaadam-Bozkurt B, Bozkurt O, Sharma H, Esposito R, ÖzoÄŸul F, Capasso R. Microbiota alteration and modulation in Alzheimer's disease by gerobiotics: The gut-health axis for a good mind. Biomed Pharmacother. 2022 Sep; 153: 113430.
Could the authors explain better in the perspectives, the benefit it could bring?
do the authors have information on whether food can also interfere with the microbiota and immunotherapy?
Could the authors better explain any side effects that immunotherapy can bring? if there are data in the literature?
Reviewer 2 Report
Review on Gut Microbiota and Immunotherapy for Alzheimer's Disease
I have completed my review on manuscript ijms-2033371, entitled, “Gut Microbiota and Immunotherapy for Alzheimer's Disease.”
Alzheimer's disease (AD) is a progressive neurodegenerative disorder that causes dementia and eventually death. There is currently no efficient treatment available to slow or stop the progression of the AD. Authors suggests that through the microbiota-gut-brain axis, the gut microbiota can modulate the host immune system in the peripheral and central nervous systems. The crosstalk between the gut microbiota, immunity, and AD immunotherapy is the focus of this article. We hypothesize that modulation of the gut microbiota caused by AD immunotherapy may contribute to the treatment's efficacy.
The topic of this review article is interesting and useful against AD. Before making a positive decision, I have some concerns and comments about the present form of the manuscript that must be addressed first.
Comments for authors
Comment 1: AD can be caused by a variety of factors. When comparing the number of factors that cause AD, the authors' description offered in the background (introduction) is insufficient to convey the information, which should be increased for new readers. Microwave was also thought to be responsible for AD. I encourage authors to add some background on this topic. The suggested article may assist authors in expanding their background knowledge and understanding the mechanisms by which the EM field interacts with and affects biological systems for various effects. The inclusion of this recent article could help to strengthen the introduction section.
Article: Microwave Radiation and the Brain: Mechanisms, Current Status, and Future Prospects. International Journal of Molecular Sciences vol. 23 (2022). [https://doi.org/10.3390/ijms23169288].
Comment 2: The gut microbiota is well known for Escherichia coli. Could you kindly provide a brief explanation of the literature review about the potential significance of E. coli in the treatment of AD? Because of how beneficial the gut microbiota is, it's probable that it plays a beneficial function in the treatment of AD.
Comment 3: Could you kindly explain how the interaction between tau protein and bacteria will help to treat AD? How this relationship functions in AD requires a brief diagrammatic representation. The authors are encouraged to make graphical representation to help readers understand their valuable work.
Comment 4: It was mentioned in the conclusion portion line number 417 that AD is a complex illness with a number of etiological variables and pathophysiological pathways. Could you please give a brief explanation of the etiological aspects involved? Because etiology encompasses a wide range of environmental factors, it is important to define this term in the conclusion section so that readers will better comprehend it.
Comment 5: It was mentioned in introduction section line 83 that the human immune system and the microbiota co-evolve and that their harmonious connection is based on the two systems' constant communication with one another. Would you kindly elaborate on this statement in the context of other neurogenerative diseases? If there was more literature on the subject with references, it would be simpler to grasp how these connections functioned not just in AD but also in other neurodegenerative conditions.
Comment 6: There are typos and inaccuracies in the paper. I strongly recommend authors to read precisely and correct the grammatical errors.
Round 2
Reviewer 2 Report
I have completed reviewing the revised version of this manuscript. The authors have addressed all of my comments and concerns in the revised version. I recommend releasing this article in its present form.